# Can Large Language Models Fix Data Annotation Errors? An Empirical Study Using Debatepedia for Query-Focused Text Summarization

**Md Tahmid Rahman Laskar[1,3], Mizanur Rahman[2,3], Israt Jahan[3],**
**Enamul Hoque[3], Jimmy Xiangji Huang[3,*]**
[1]Dialpad Canada Inc., [2]Royal Bank of Canada, [3]York University[†]
Toronto, Ontario, Canada
`{tahmid20,mizanurr,israt18,enamulh,jhuang}@yorku.ca`

## Abstract

Debatepedia is a publicly available dataset consisting of arguments and counter-arguments on controversial topics that has been widely used for the single-document query-focused abstractive summarization task in recent years. However, it has been recently found that this dataset is limited by noise and even most queries in this dataset do not have any relevance to the respective document. To this end, this paper aims to study whether large language models (LLMs) can be utilized to clean the Debatepedia dataset to make it suitable for query-focused abstractive summarization. More specifically, we harness the language generation capabilities of two LLMs, namely, ChatGPT[1], and PaLM[2] to regenerate its queries. Based on our experiments, we find that only fixing the queries in Debatepedia via LLMs may not be useful. However, leveraging a rule-based approach via filtering out noisy instances followed by query regeneration using LLMs for the sampled instances may ensure a higher quality version of this dataset suitable for the development of more generalized query-focused text summarization models.

## 1 Introduction

Text summarization is a natural language processing technique that involves generating a concise and coherent summary of a longer piece of text while preserving its most important information (Yao et al., 2017). Query-focused text summarization is a specific type of summarization that generates a summary of the given text that is focused on answering a specific question (Laskar et al., 2020c) or addressing a particular topic, rather than providing a general overview of the text. (Baumel et al., 2018; Goodwin et al., 2020; Su et al., 2020; Xu and Lapata, 2021; Laskar et al., 2020a,b, 2022).

One widely used dataset for this task is the Debatepedia dataset that consists of arguments and counter-arguments on conversational topics (Nema et al., 2017). The query-focused summarization of argumentative text is a challenging task that has gained increasing attention in recent years due to its potential applications in various domains, such as policy-making, journalism, and legal reasoning.

However, it has been recently found that the quality of the Debatepedia dataset that is widely used for this task is limited by noise, with many of the queries in this dataset having no relevance with the source document (Laskar et al., 2022). Since Debatepedia is a rich source of argumentative text on controversial topics that can serve as a valuable resource for the development and evaluation of summarization models, in this paper, we present a novel methodology to clean the Debatepedia dataset via re-annotation of its queries to make it a useful resource for query-focused abstractive summarization. Our data annotation approach leverages large pre-trained language models (Devlin et al., 2018; Brown et al., 2020; Ouyang et al., 2022), such as ChatGPT (OpenAI, 2023) and PaLM-2 (Anil et al., 2023), that have demonstrated impressive capability of generating fluent and coherent text (Laskar et al., 2023a). Using these LLMs, we regenerate the queries in the Debatepedia dataset that are more likely to have no relevance to the document and the summary. More specifically, this paper aims to investigate whether LLMs can be utilized to fix the existing issues in the Debatepedia dataset. Our extensive experiments show that utilizing rule-based filtering to eradicate noisy instances alongside leveraging the generative power of LLMs to regenerate the irrelevant queries leads to performance improvement in terms of both query relevance and summary generation quality. We will make this LLM-annotated cleaned version of Debatepedia publicly available[3].

---

*Contact Author.
[†]All work being done at York University.

[1]`https://openai.com/blog/chatgpt/`
[2]`https://ai.google/discover/palm2`

[3]`https://github.com/tahmedge/CQSUMDP`

## 2 Related Work

Query-focused text summarization using neural models has gained increasing attention in recent years (Baumel et al., 2018; Laskar et al., 2022). The recent success of transformer-based models (Vaswani et al., 2017; Lewis et al., 2019; Raffel et al., 2019; Zhang et al., 2019) on generic abstractive summarization has also inspired researchers to utilize such models for query-based abstractive summarization (Goodwin et al., 2020; Vig et al., 2021; Laskar et al., 2020a,b, 2022), leading to state-of-the-art performance in benchmark query-based summarization datasets, such as DUC (Feigenblat et al., 2017; Xu and Lapata, 2020), AQuaMuSe (Kulkarni et al., 2020), QMSum (Zhong et al., 2021), Debatepedia (Nema et al., 2017), etc.

Among the datasets mentioned above, one notable exception is the Debatepedia dataset since it requires generating summaries from a document containing argumentative text (i.e., arguments and counter-arguments). However, it has been found recently that many samples in the Debatepedia dataset are not actually query oriented while models that are trained without considering the query relevance could achieve almost similar performance as the query-focused summarization models (Laskar et al., 2022). Thus, there remains a scarcity of datasets specifically tailored to generate query-focused summaries of argumentative texts.

Though some studies (Abdullah and Chali, 2020) have attempted to generate the queries in generic summarization datasets (e.g., CNNDM (Nallapati et al., 2016)), we find that these queries are generated by directly extracting words from the reference summaries, leading to unexpected access to the keywords in the reference summaries for the summarization models. LLMs have received a lot of attention recently due to their impressive language generation capability – ensuring high fluency, coherence, and grammatical correctness on the generated texts (Laskar et al., 2023a; Qin et al., 2023; Bang et al., 2023; Yang et al., 2023; Wang et al., 2023; Kocoń et al., 2023). More importantly, ChatGPT like LLMs also demonstrated impressive capability for data annotation (Wang et al., 2021; Ding et al., 2022; Gilardi et al., 2023). To this end, in this paper, we study how to fix the queries in Debatepedia using LLMs to construct a cleaned version of the dataset to make it suitable for query-focused summarization of argumentative texts.

## 3 Our Annotation Methodology

Debatepedia is a publicly available dataset of arguments and counter-arguments on debate topics, proposed by Nema et al. (Nema et al., 2017). It contains about 13K query-document-summary pairs. The average number of words per document, summary, and query in the Debatepedia dataset is 66.4, 11.16, and 9.97, respectively. The dataset covers a wide range of topics, such as politics, sports, and technology, and has been extensively used in recent years to build query-based summarization models for argumentative text (Laskar et al., 2022). However, the quality of Debatepedia as a dataset for query-based summarization has lots of limitations (see Table 5 in Appendix A.1 for some examples), as it has been found recently that many queries in this dataset are not relevant to the document (Laskar et al., 2022). To address these limitations, we propose a methodology for cleaning the Debatepedia dataset via leveraging two popular LLMs: ChatGPT (Laskar et al., 2023a) and PaLM-2 (Anil et al., 2023), as annotators. In this regard, we initially explored various techniques to identify how to effectively sample the noisy instances, and subsequently, we regenerated the queries for the sampled instances. We denote our **C**hatGPT and **P**aLM annotated versions of **D**ebate**p**edia (DP) for **Q**uery Focused Abstractive **Sum**marization as the **CQSumDP** and the **PQSumDP**, respectively.

### 3.1 Data Sampling

We explore two approaches for data sampling. In one approach, we study whether only fixing the queries in the Debatepedia dataset via leveraging LLMs for query regeneration could address the issues in the Debatepedia dataset or not. For this purpose, we ask LLMs to identify the instances in the Debatepedia dataset where the queries seemed irrelevant. In our other approach, we first sample data instances based on some filtering rules by excluding instances that are less relevant for query-focused summarization, and then we ask LLMs to re-generate the queries from these sampled instances where the queries looked irrelevant. Our prompt for data sampling using LLMs is shown in Table 1(a). Below, we describe these approaches.

**(i) LLM-based Data Sampling without Filtering:** In this approach, we use the full Debatepedia dataset to find the irrelevant queries using LLMs. For this purpose, we provide each instance of Debatepedia to the LLMs to determine if the query is

| (a) **Prompt: Data Sampling for Query Regeneration** | (b) **Prompt: Regenerating the Sampled Queries** |
|---|---|
| *Below, we provide a query, a document, and the query-focused summary of the given document. Identify whether the query is relevant to the summary? Answer as either yes or no.*
*Query: [QUERY]*
*Document: [DOCUMENT]*
*Summary: [SUMMARY]* | *A document along with its summary are given below. Write down the most reasonable query relevant to this document-summary pair?*
*Document: [DOCUMENT]*
*Summary: [SUMMARY]* |

Table 1: Prompts for LLMs: (a) data sampling for query regeneration, and (b) regenerating the sampled queries.

relevant to the document/summary. However, we find a significant difference between LLMs in this task. While PaLM-2 only identifies 659 queries as irrelevant (612/19/28 in train/valid/test sets, respectively), ChatGPT identifies 6435 queries as irrelevant (5697/316/422 in train/valid/test sets, respectively), out of 13719 samples.

**(ii) LLM-based Data Sampling via Filtering:** In this approach, instead of cleaning the Debatepedia dataset by only fixing the queries, we also follow some rules to first filter out some irrelevant instances from the dataset to address the existing limitations in Debatepedia (Laskar et al., 2022), such as smaller-sized documents, close-ended questions, etc. Since for the smaller-sized documents, the reference summaries are mainly the overall generic summary of the document where the additional query does not help, we aim to exclude smaller-sized documents to ensure that the reference summaries are more query-focused. This also helps us to address the noisy scenario in the dataset when the reference summary length is longer than the document length. Based on manual analysis, we find that a minimum length of 75 words for each selected document at least ensures a document where the query could play a role in the summary generation. To also address the issue of short summaries that looked like answers to closed-ended questions, we exclude instances where the length of the summary is shorter than 5 words. This helps us to clean the dataset in a way such that instead of having a dataset with close-ended questions and short answers, we propose a dataset consisting of concise but coherent summaries. This results in a filtered version of the dataset which is smaller in size, consisting of 5291/309/405 instances, in comparison to the original dataset[4] containing 12000/719/1000 instances, in train/valid/test sets, respectively. We also find that ChatGPT and PaLM-2 identified 2171/120/145 and 218/6/6 queries as irrelevant in

the training, validation, and test sets, respectively.

Below, we demonstrate how we utilize LLMs for query regeneration.

### 3.2 Using LLM for Query Regeneration

We concatenate the document and the reference summary together and give as input to the LLMs for query regeneration. Our sample prompt for this task can be found in Table 1(b). While we could ask LLMs to generate both the query and the query-based summary by only giving the document in the input prompt, we did not do so since it is found that LLMs like ChatGPT tend to generate longer summaries (Laskar et al., 2023a; Qin et al., 2023) while the resulting dataset could become a fully synthetic dataset. Thus, we use both the document and the summary as input and only regenerate the queries while keeping the original reference summaries intact. We find that the regenerated queries using ChatGPT and PaLM-2 only have 15.2% and 11.4% word overlaps, respectively, with the gold summaries, in comparison to the 10.6% word overlaps in the original Debatepedia dataset.

### 4 Experimental Results

In this section, we present our experimental findings. We denote the version of our dataset where we did not apply any filtering as the `unfiltered` version, whereas we denote the version of our dataset where we also utilize filtering while sampling data instances based on applying some rules as the `filtered` version. For ChatGPT, we use the `gpt-3.5-turbo-0301`[5] model; while for PaLM-2, we use the `text-bison@001`[6] model. We fine-tune the following models to benchmark the performance in our re-annotated versions of Debatepedia since these models achieved impressive performance in query-focused abstractive summarization

---

[4]https://github.com/PrekshaNema25/
DiverstiyBasedAttentionMechanism/tree/master

[5]https://platform.openai.com/docs/models/
gpt-3-5
[6]https://cloud.google.com/vertex-ai/docs/
generative-ai/learn/models

| Model | Training | Evaluation | ROUGE-1 | ROUGE-2 | ROUGE-L | BERTScore |
|---|---|---|---|---|---|---|
| BART-Base | Original Debatepedia | MS-MARCO | 37.8 | 20.0 | 34.3 | 68.8 |
| Pegasus-Base | Original Debatepedia | MS-MARCO | 31.2 | 14.9 | 27.7 | 59.9 |
| T5-Base | Original Debatepedia | MS-MARCO | 45.1 | 26.9 | 40.9 | 71.9 |
| BART-Base | CQSumDP (filtered) | MS-MARCO | 42.3 | 24.2 | 38.2 | 70.5 |
| Pegasus-Base | CQSumDP (filtered) | MS-MARCO | 47.1 | 31.4 | 43.3 | 69.0 |
| T5-Base | CQSumDP (filtered) | MS-MARCO | **47.6** | **29.4** | **43.0** | **73.1** |
| BART-Base | PQSumDP (filtered) | MS-MARCO | 41.4 | 23.3 | 37.5 | 70.6 |
| Pegasus-Base | PQSumDP (filtered) | MS-MARCO | 44.7 | 28.2 | 40.7 | 68.7 |
| T5-Base | PQSumDP (filtered) | MS-MARCO | 47.3 | 29.0 | 42.8 | 73.0 |
| BART-Base | CQSumDP (unfiltered) | MS-MARCO | 38.4 | 20.6 | 34.8 | 68.4 |
| Pegasus-Base | CQSumDP (unfiltered) | MS-MARCO | 43.7 | 27.3 | 40.0 | 67.7 |
| T5-Base | CQSumDP (unfiltered) | MS-MARCO | 44.7 | 26.5 | 40.4 | 71.9 |
| BART-Base | PQSumDP (unfiltered) | MS-MARCO | 39.1 | 21.4 | 35.5 | 69.1 |
| Pegasus-Base | PQSumDP (unfiltered) | MS-MARCO | 40.1 | 23.1 | 36.3 | 66.2 |
| T5-Base | PQSumDP (unfiltered) | MS-MARCO | 44.4 | 25.6 | 40.0 | 72.3 |

Table 2: Performance of different models on MS-MARCO when trained on respective versions of Debatepedia (DP).

| Model | Dataset | ROUGE-1 | ROUGE-2 | ROUGE-L | BERTScore |
|---|---|---|---|---|---|
| BART-Base | CQSumDP (filtered) | **39.6** | **22.1** | **36.6** | **70.8** |
| Pegasus-Base | CQSumDP (filtered) | 31.5 | 13.9 | 28.4 | 66.8 |
| T5-Base | CQSumDP (filtered) | 31.3 | 13.2 | 28.6 | 67.1 |
| BART-Base | PQSumDP (filtered) | 37.8 | 21.,2 | 35.6 | 70.3 |
| Pegasus-Base | PQSumDP (filtered) | 27.1 | 10.8 | 24.9 | 64.9 |
| T5-Base | PQSumDP (filtered) | 28.0 | 10.6 | 25.4 | 65.6 |
| BART-Base | CQSumDP (unfiltered) | **41.6** | **23.4** | **39.1** | **72.3** |
| Pegasus-Base | CQSumDP (unfiltered) | 33.4 | 15.8 | 30.6 | 68.3 |
| T5-Base | CQSumDP (unfiltered) | 33.9 | 15.1 | 31.0 | 68.6 |
| BART-Base | PQSumDP (unfiltered) | 39.8 | 22.2 | 37.2 | 71.7 |
| Pegasus-Base | PQSumDP (unfiltered) | 29.2 | 12.5 | 26.6 | 66.3 |
| T5-Base | PQSumDP (unfiltered) | 30.2 | 12.6 | 27.6 | 66.9 |

Table 3: Performance of different models on various versions of Debatepedia.

in recent years (Laskar et al., 2022; Goodwin et al., 2020): (i) BART (Lewis et al., 2019), (ii) T5 (Raffel et al., 2019), and (iii) Pegasus (Zhang et al., 2019). Similar to prior work on query-focused text summarization (Laskar et al., 2022), we concatenate the query with the document and give as input to these models to generate the query-focused summaries. For all models, we use their respective Base versions from HuggingFace (Wolf et al., 2019) (see Appendix A.3 for more details). For all models, the results are evaluated using ROUGE-1, ROUGE-2, ROUGE-L, and BERTScore. For BERTScore, we use the DeBERTa-xlarge-mnli (He et al.) model.

## 4.1 Effectiveness of LLMs for Data Cleaning

In this section, to investigate the effectiveness of using LLMs for data cleaning, we evaluate the performance of models trained on different LLM-annotated versions of the Debatepedia dataset in an out-of-domain dataset for the query-focused abstractive summarization task. This is done to ensure that all models are evaluated on the same evaluation set. In this regard, we use the development set of the QA-NLG version of the MS-MARCO (Wang et al., 2018) dataset (12467 samples). We follow the similar settings of Laskar et al. (2022) by only

considering the gold passage as the source document, and after combining the passage with the query we give the concatenated text as input to the models. The results of all three models (BART, T5, Pegasus) on MS-MARCO that are fine-tuned on the respective versions of Debatepedia are shown in Table 2. We observe based on our experimental results that the domain generalization performance is much better when the CQSumDP/PQSumDP versions of the Debatepedia dataset are used in comparison using the Original Debatepedia dataset. While comparing ChatGPT and PaLM as data annotators, we observe that models trained on CQSumDP perform better than PQSumDP. Moreover, we find that models trained on the filtered version obtain better performance (with T5-Base achieving the best result), indicating the importance of cleaning the Debatepedia dataset by excluding noisy instances, alongside utilizing LLM-generated queries.

**Qualitative Evaluation of Model Generated Summaries:** We sample 10 summaries generated by each model (BART, T5, Pegasus) on the MS-MARCO dataset to conduct human evaluations for our best-performing approach, the CQSumDP (filtered version), and the baseline Original Debatepedia. In our human evaluation, we ask humans

to score between 1 to 5 for the factual consistency and the coherence of the summaries generated by different models for the given queries. The average coherence and factual consistency scores for models trained on CQSumDP (filtered) are **3.4** and **3.3**, respectively; in comparison to the average coherence and factual consistency scores of **3** and **2.6**, respectively, for the Original Debatepedia. This further establishes the effectiveness of using LLMs as annotators to make a more suitable dataset for query-focused text summarization.

**Qualitative Evaluation of LLM Generated Queries:** We sample 100 instances and ask three human evaluators to choose between the ChatGPT and PaLM-generated (see Appendix A.4 for some examples) queries that they prefer based on the conciseness and the relevancy of the query. We find that in 66% cases (via majority voting), the ChatGPT-generated queries were preferred.

**LLM as Query Relevancy Classifier:** To measure the capability of LLMs in classifying whether the query is relevant to the document/summary, we sample 100 instances and evaluate using three human evaluators to find if they also agree with the classification done by LLMs. We find based on majority voting that the precision for the classification task for PaLM-2 is 75%, while for ChatGPT is 63%. This trend of PaLM-2 outperforming Chat-GPT in discriminative tasks (e.g., classification) while being underperformed in generative tasks is also observed in recent studies (Jahan et al., 2023).

**Ablation Studies:** To further investigate the usefulness of LLM-generated queries, we conduct the following ablation tests using the best-performing model, **T5-base**, on MS-MARCO (see Table 4).

*(i) Remove LLM-generated Query:* Here, we evaluate the performance of the T5 model by fine-tuning it on the filtered version of Debatepedia without incorporating any query relevance. We find based on the average score across different metrics that the performance for T5 is dropped by 9.53% on average, in comparison to the T5 model fine-tuned on the CQSumDP (filtered) dataset.

*(ii) Replace LLM-generated Query:* Here, we evaluate the performance by fine-tuning T5 using the original query instead of the LLM-generated query in the filtered version of Debatepedia. Based on the average scores achieved by the T5 model, the performance is dropped by 3.57% on average, compared to T5 fine-tuned on CQSumDP (filtered).

| Type | ROUGE-1 | ROUGE-2 | ROUGE-L | BERTScore |
|------|---------|---------|---------|-----------|
| **ChatGPT Generated Query** | **47.6** | **29.4** | **43.0** | **73.1** |
| *Without Query Relevance* | 42.2 | 23.7 | 38.0 | 70.8 |
| *Original Query* | 45.7 | 29.2 | 41.6 | 69.7 |

Table 4: Ablation test result for the T5-Base model fine-tuned on Debatepedia (filtered) and evaluated on MS-MARCO.

**Cost and Time Efficiency:** Recently, it was found that LLMs could significantly reduce the labeling cost without sacrificing the model's performance much, making it possible to train models on larger datasets without the need for human labeling (Wang et al., 2021; Ding et al., 2022; Liu et al., 2023). In this work, we observe that Chat-GPT/PaLM APIs could generate about 15 queries on average per minute, which should be much faster than using human annotators, since humans may need some time to come up with the most effective query for the given document-summary pairs. This makes LLMs to be more effective for annotation.

### 4.2 Performance Benchmarking on Different Versions of Debatepedia

In this section, we benchmark the performance of various LLM-annotated versions of the Debatepedia dataset. We present our results in Table 3 to find that all three models perform better in the CQSumDP dataset in comparison to their performance on the PQSumDP. This gives further indication that the queries generated by ChatGPT are more helpful in improving the model performance. While comparing between different models, we find that in both the filtered and the unfiltered versions, the best performance is achieved by the BART model.

## 5 Conclusions and Future Work

In this paper, we study how to effectively leverage LLMs to construct a cleaned version of the Debatepedia dataset to address the existing limitations in this dataset in order to make it suitable for query-focused text summarization. Based on extensive experiments and evaluation, we demonstrate that our proposed data re-annotation approach using LLMs (especially ChatGPT) results in a cleaner version of Debatepedia that is found to be more effective for the query-focused summarization task in comparison to the original dataset. In the future, we will explore whether few-shot examples with LLMs lead to better performance. Our re-annotated versions of Debatepedia will also be made publicly available here: https://github.com/tahmedge/CQSUMDP.

## 6 Limitations

ChatGPT (GPT-3.5) and PaLM models are continuously upgraded by OpenAI and Google. Thus, it may not be possible to reproduce the same queries using these models. However, this also mimics the real-world scenario as different human annotators may write different queries (e.g., in many text summarization datasets, there can be multiple gold reference summaries written by different human annotators). However, similar to the work of Guo et al. (2023), we also notice that this difference is very small. Therefore, we also generate only one query for each example. Though a new version of ChatGPT called GPT-4[7] has been recently released which may generate more powerful queries, in this work, we did not utilize GPT-4 as it is quite expensive to use than the original ChatGPT (i.e., GPT-3.5) while being significantly slower. Nonetheless, future work may compare with other more powerful LLMs (including GPT-4) for data annotation.

## 7 Ethics Statement

Since this paper only utilizes LLMs to generate the queries for the given document-summary pairs, it does not lead to any unwanted biases or ethical concerns. However, all the responses generated by ChatGPT and PaLM are still manually checked by the authors to ensure that the LLM-generated queries in the cleaned version of the dataset do not pose any ethical concerns or unwanted biases. Only a publicly available academic dataset was used that did not require any licensing. Thus, no personally identifiable information has been used while utilizing LLMs to fix the queries in the Debatepedia dataset. All the human evaluators were also paid above the minimum wage.

## Acknowledgements

We would like to thank all the anonymous reviewers and the area chairs for their excellent review comments. This research is supported in part by the Natural Sciences and Engineering Research Council (NSERC) of Canada, the York Research Chairs (YRC) program, and the Generic Research Fund of York University. We also acknowledge Compute Canada for providing us with the computing resources. Jimmy Huang (*jhuang@yorku.ca*) is the contact author of this paper.

---

[7] https://openai.com/research/gpt-4

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

# A  Appendix

## A.1  Debatepedia Dataset Limitations

Based on a randomly sampled 100 instances, it has been found in a recent study (Laskar et al., 2022, 2023b) that:

- 52% of the queries in this dataset have no relevance to the documents or the summaries, as demonstrated in Table 5.

- 70% of the queries are close-ended (i.e., Yes/No type) questions (see Example 4 in Table 5).

- Though, many queries in this dataset are relevant to the documents but the summaries are more of generic due to shorter document length. The average length of the document in this dataset is only 66.4 words.

In addition, many instances in this dataset only contain one word summary (see Example 2 in Table 5) for a given query that appears both in the training and evaluation sets, which may also help the model to memorize such words for similar queries during the training phase. These issues may lead to an unexpected increase in the ROUGE score when the model starts learning to reproduce those words in the summary during the evaluation phase. Furthermore, we also find some instances where the length of the summary is longer than the document length, which usually happens in short documents (see Example 3 in Table 5).

## A.2  Example Prompt for Query Generation

One example prompt to re-generate the query using LLMs is shown in Figure 1.

## A.3  Experimental Details

### A.3.1  Models

To evaluate the effectiveness of our ChatGPT annotated CQSumDP and PaLM annotated PQSumDP datasets, we fine-tune some state-of-the-art pre-trained sequence to sequence models (Lewis et al., 2019; Raffel et al., 2019; Zhang et al., 2019; Goodwin et al., 2020). For this purpose, we concatenate the query with the document and give as input to these models to generate the query-focused abstractive summaries as this approach has shown impressive performance in the query-focused abstractive summarization task recently (Laskar et al., 2022). We describe these models below:

| |
|---|
| *Example 1: Query having no relevance with the document and the summary.* |
| **Query:** Does an MBA enhance leadership skills? |
| **Document:** Business schools might improve your quantitative presentation and communication skills. It might but get you thinking about ethical and strategy. But two years of case studies aren't go to turn you into a leader if you weren't died one. There's no learning charisma persuasiveness elegance or gut instinct. |
| **Reference Summary:** PhD will not improve cm factors of leaders. |
| *Example 2: One word summary having no relevance with the query or document.* |
| **Query:** Education : do child benefit from watching tv? |
| **Document:** by watching news child can learn about geography politics advances in science – everything simply and later explained . furthermore child learn about real-life situation that happens on everyday basis which will benefit them in the future. |
| **Reference Summary:** News. |
| *Example 3: The length of the summary is longer than the document with the query being irrelevant.* |
| **Query:** activists : where do the keys activists and organizations stand ? |
| **Document:** see an analyses of the article ... |
| **Reference Summary:** philip martin of berkeley davis and michael teitelbaum the mirage of mexican guest workers nov/dec # foreign affairs . |
| *Example 4: More of a close-ended question.* |
| **Query:** friendships : does twitter harms relationships ? |
| **Document:** twitter helps those stay in touches no matter how far they may be from each other . |
| **Reference Summary:** long-distance friendships . |

Table 5: Some examples demonstrating the limitations in the Debatepedia dataset.

**BART (Bidirectional and Auto-Regressive Transformer):** BART (Lewis et al., 2019) is a pre-trained sequence-to-sequence model based on the encoder-decoder architecture that was pre-trained on a large amount of diverse text data using the denoising auto-encoding technique to recover the original form of a corrupted document. The pre-training involved various objectives such as rotating the document, permuting sentences, infilling text, masking tokens, and deleting tokens. We use the pre-trained BART model since fine-tuning this model was found to be very effective in abstractive summarization (Laskar et al., 2022).

**T5 (Text-to-Text Transfer Transformer):** The T5 model (Raffel et al., 2019) is a transformer-based model based on the BERT architecture. However, unlike traditional BERT models that classify input text into a specific category, the T5 model treats all tasks such as text classification, question answering, neural machine translation, and text summarization as a sequence-to-sequence problem using various pre-training objectives. After pre-training, the model is fine-tuned on many down-stream tasks, achieving impressive performance across various datasets including summarization.

**Pegasus (Pre-training with Extracted Gap-sentences for Abstractive Summarization):** Pegasus (Zhang et al., 2019) is a transformer-based pre-trained encoder-decoder model for abstractive summarization. Its pre-training objective involves generating summary like text from an input document. To achieve this, the PEGASUS model first selects and masks some sentences from the input document(s). It then concatenates these selected sentences to create a pseudo-summary. The model uses different approaches to select these sentences, such as randomly selecting a certain number of sentences, selecting the first few sentences, or computing the ROUGE-1 score between each sentence and the rest of the document to choose the top-scoring sentences. This pseudo-summary is then used for self-supervised learning. By pre-training on large datasets using this approach, the model achieves impressive fine-tuning performance on downstream summarization datasets.

### A.3.2 Implementation

We use the HuggingFace[8] (Wolf et al., 2019)library to implement the baseline models for performance evaluation. Similar to the prior work, we concate-

---

[8]https://huggingface.co/

Figure 1: Example Input to LLMs for Query Generation.

nated the query with the document to give as input to the pre-trained baselines (i.e., BART, Pegasus, T5). The pre-trained model is then fine-tuned using 4 NVIDIA V100 GPUs. The training batch size for BART was set to 16, while it was set to 4 for Pegasus and T5. The other hyperparameters were similar for all models, with the learning rate being set to $2e-3$ and the maximum input (i.e., the concatenated query and document) sequence length being 150 tokens. The minimum and the maximum target (i.e., the generated summary) sequence lengths were 5 and 25, respectively. A total of 10 epochs were run to fine-tune the pre-trained summarization models. We computed the ROUGE (Lin, 2004) scores in terms of ROUGE-1, ROUGE-2, and ROUGE-L using the *Evaluate*[9] library to compare the performance of different models on the respective test set. As noted earlier, for ChatGPT, we use the `gpt-3.5-turbo-0301` model; while for PaLM, we use the `text-bison@001`model.

### A.4 Qualitative Analysis of the Annotated Data

In this section, we do some qualitative analyses between the queries in the Original Debatepedia dataset as well as the queries generated using LLMs in our proposed CQSumDP and PQSumDP versions of the Debatepedia dataset. For our analysis, we collect a set of 3 samples from this dataset and present them in Table 6. While comparing between the queries in the first example in the table, we find that the original query is just one word length and very ambiguous, while the ChatGPT generated query is more descriptive and more relevant to both the document and the summary. For the second ex-

ample, we find that even though the original query is descriptive, it does not have any relevance to the generated summary. Whereas both the ChatGPT and PaLM generated queries are very relevant to both the document and the summary (in this example, PaLM generated query is more descriptive). For the third example, we find that the original query is related to "entrepreneurs". However, the document is about "product managers", not "entrepreneurs". Meanwhile, the ChatGPT and PaLM generated queries are also very relevant to the document and both LLM-generated queries are the same. This analysis further demonstrates the relevance of our LLM-generated query in comparison to the original query in Debatepedia.

---

[9]https://huggingface.co/spaces/evaluate-metric/rouge

| # | Original Query | ChatGPT Query | PaLM Query | Source Document | Gold Summary |
|---|---|---|---|---|---|
| 1. | military : | What actions did the government take to improve the situation for U.S. troops and veterans? | What did obama do for veterans ? | provided better body armor to our troops . provided the department of veterans affairs ( va ) with more than $ # . # billion to improve services to america s veterans . ended media blackout on war casualties and the return of fallen soldiers to dover afb . announced creation of a joint virtual lifetime electronic record for members of the u.s. armed forces to improve quality of medical care . ended the previous stop-loss policy that kept soldiers in iraq/afghanistan longer than their enlistment date . signed the veterans health care budget reform and transparency act authorizing advance appropriations for the department of veterans affairs by providing two-fiscal year budget authority thus enabling better medical care for veterans . endorsed by the american legion american veterans blinded veter ... ans association | improved services benefits and respect for troops . |
| 2. | we economy : has wto benefited the economy of the united states ? | Has NAFTA caused job losses in the U.S? | What does the author say about the effect of NAFTA on American wages? | " nafta and job losses " . cyril morong ( PhD ) the wall street journal may # # - " did nafta cause the u.s. to lose so many jobs [ citing figures provided in the range of # million and # # ] especially high-paying manufacture jobs ? probably not . i say probably since causality in any social science ( economics included ) is difficult to prove since so many factors change so quickly in the real world . but if many high-paying manufacture jobs were lost it took many years until after nafta went into effect before they were ... but what about manufacture jobs ? we had just about # million in # . it actually rose to # . # million in # and was at # . # in #. | nafta has decreased the number of american job |
| 3. | entrepreneurs: does an mba help entrepreneurs ? | Is an MBA necessary for product managers? | Is an MBA necessary for product managers? | christopher cummings . " is an mba necessary for product managers ? " product management meet pop culture . february # # : " hindsight . looking back the brass tacks of my mba experience were about the basics of management economics and business strategy . could that have been picked up on the job ? maybe . [ ... ] however the more important throughline of the experience relates to critical thinking perspective and learning when to lead and when to follow . [ ... ] on the job especially as a young pm it can be easy to lose perspective to miss the forest for the trees . at the time i was definitely into the plate-spinning the go-go-go the tactics and day-to-day . no time to think ; just keep moving . [ ... ] the mba experience | mba teach strategy plan not just tactics |

Table 6: Comparisons between the original queries and the LLM-generated queries in some samples of the Debatepedia dataset. Note that the personally identifiable information in this dataset is anonymized with the # token.