# OpenReview forum: "Can Large Language Models Fix Data Annotation Errors? An Empirical Study Using Debatepedia for Query-Focused Text Summarization"
_EMNLP/2023/Conference — EMNLP 2023 Findings_

### Official Review · Reviewer_UJ5T · 2023-08-04

**Typos Grammar Style And Presentation Improvements:** Tables 1 and 2 are hard to read witho…
**Soundness:** 4

**Excitement:**

3: Ambivalent: It has merits (e.g., it reports state-of-the-art results, the idea is nice), but there are key weaknesses (e.g., it describes incremental work), and it can significantly benefit from another round of revision. However, I won't object to accepting it if my co-reviewers champion it.

**Paper Topic And Main Contributions:**

The paper is about using LLMs (ChatGPT and PaLM) to improve the Debatepedia dataset to make it more suited for query-focused abstractive summarization. Recent work has found that many of the queries in the dataset have no relevance with the source document, and the paper proposes two approaches for data sampling to identify and regenerate these queries. Models fine-tuned on the LLM-annotated dataset usually perform better than those trained on.

**Questions For The Authors:**

1. Could you provide some additional insights into the qualitative differences of summaries trained with and without LLM-annotated data?

**Reasons To Accept:**

1. The paper addresses a well-known issue with an existing dataset and shows how to efficiently leverage LLMs to correct it. The approach is simple and does improve performance.

**Reasons To Reject:**

1. The paper is rather narrow in scope, only applying the method to a single dataset, while the approach could have larger implications to other query-focused datasets or noisy datasets in general.
2. The paper only briefly mentions a human evaluation (a bit more discussed in the appendix) of the queries produced. A comparison of the summaries produced with and without the LLM-annotated data would be insightful.

**Reproducibility:**

4: Could mostly reproduce the results, but there may be some variation because of sample variance or minor variations in their interpretation of the protocol or method.

**Reviewer Confidence:**

4: Quite sure. I tried to check the important points carefully. It's unlikely, though conceivable, that I missed something that should affect my ratings.

---

> ### Author Rebuttal · Authors · 2023-08-29
>
> We would like to thank you for thoroughly reading the paper and providing constructive feedback. We have carefully read your reviews and prepared our responses accordingly to address your concerns. Our response to your comments is stated below.
>
> **Response to Reasons to Reject:**
>
> **The paper is rather narrow in scope, only applying the method to a single dataset, while the approach could have larger implications to other query-focused datasets or noisy datasets in general:** While we agree that the paper is quite narrow in score, this is also the primary reason for submitting to the short paper track as the early-stage findings of this paper are more relevant to the requirements of short papers. As the Debatepedia dataset has existing limitations in terms of query relevance, we believe that the solution proposed by us will not only be useful for researchers working in this area but can also be extended to other datasets and tasks.
>
> **Questions for the Authors:**
>
> We initially reported the results based on only the ROUGE metric as this is mostly done when the results are reported for Debatepedia [1, 2]. Since we conducted extensive experiments across various models (BART, T5, PEGASUS) in different scenarios, filtered and unfiltered versions, on both Debatepedia for in-domain evaluation as well as MS-MARCO for domain generalization, conducting human evaluation on the outputs of all models will require significant time. Thus, we utilize GPT-4 to conduct human-like qualitative evaluation due to its impressive performance as evaluators across various tasks, including state-of-the-art performance on the summarization task as human-like evaluators [3].
>
> In our human-like qualitative evaluation using GPT-4, we ask GPT-4 to score between 1 to 5 for the factual consistency and the coherence of the summaries generated by different models, as generating a factually correct summary as well as maintaining coherence during summary generation are two of the important metrics that may depend on the given queries.
>
> We sample 10 summaries generated by each model on the MS-MARCO dataset and compute the average score for the filtered version that is based on LLM-generated queries and the unfiltered version that is based on the original queries. Our results are stated below. We find that the factual consistency and the coherence score of the models are much better in the filtered version in comparison to the unfiltered original version:
>
> CQSUMDP (filtered):
> - Coherence: 2.87
> - Factual Consistency: 3.00
>
> PQSUMDP (filtered):
> - Coherence: 3.30
> - Factual Consistency: 2.93
>
> Original Debatepedia (unfiltered):
> - Coherence: 1.33
> - Factual Consistency: 0.98
>
> References:
>
> [1] Preksha Nema, Mitesh M. Khapra, Anirban Laha, and Balaraman Ravindran. 2017. Diversity driven attention model for query-based abstractive summarization. In Proceedings of the 55th Annual Meeting of the Association for Computational Linguistics (Volume 1: Long Papers), pages 1063–1072, Vancouver, Canada. Association for Computational Linguistics.
>
> [2] Md Tahmid Rahman Laskar, Enamul Hoque, and Jimmy Xiangji Huang. 2022. Domain Adaptation with Pre-trained Transformers for Query-Focused Abstractive Text Summarization. Computational Linguistics, 48(2):279–320.
>
> [3] Liu, Yang, Dan Iter, Yichong Xu, Shuohang Wang, Ruochen Xu, and Chenguang Zhu. "Gpteval: Nlg evaluation using gpt-4 with better human alignment." arXiv preprint arXiv:2303.16634 (2023).

---

### Official Review · Reviewer_2X9Y · 2023-08-05

**Soundness:** 3

**Excitement:**

3: Ambivalent: It has merits (e.g., it reports state-of-the-art results, the idea is nice), but there are key weaknesses (e.g., it describes incremental work), and it can significantly benefit from another round of revision. However, I won't object to accepting it if my co-reviewers champion it.

**Paper Topic And Main Contributions:**

This paper investigates the task setting of leveraging large language models (LLMs) to clean noisy datasets. The authors use two approaches mostly for this purpose: (1) a rule-based method for filtering out low-quality examples; (2) using LLMs to rewrite the queries for query-focused text summarization. Using Debatepedia as the example dataset, the authors show that the dataset cleaned by LLMs is able to help the models to achieve better performance on the dataset that is used for fine-tuning and other related datasets under automatic evaluation.

**Questions For The Authors:**

What are the inter-annotator agreement and the specific setting of the human evaluation you conducted for comparing the quality of queries written by ChatGPT and PaLM?

**Reasons To Accept:**

This paper provides a case study of leveraging LLMs for cleanling noisy datasets. This idea in general can have great potential.

**Reasons To Reject:**

The experimental/evaluation setting of this paper cannot sufficiently support the claim of this work. Specifically, the authors give two reasons for arguing the effectiveness of LLM-based data cleaning:
- The models that are trained on the cleaned dataset can achieve better performance in automatic evaluation on the *same* dataset that is used for model training. I'm not sure why this can support the claim that the LLM-cleaned dataset is better, since the performance of the model trained on a dataset is not a direct measurement of the quality of the same dataset.
- The models trained on the LLM-cleaned dataset show better performance on other datasets which demonstrates better generalization ability.  This is a more convincing point than the first one, however, the improvement seems to be marginal under automatic evaluation and the authors did not perform human evaluation to compare the performance of the models trained on the original dataset and the cleaned dataset.
- While the authors provide a human evaluation for comparing the quality of generated queries from ChatGPT and PaLM, they did not provide a similar study for comparing the query quality of the original dataset and the LLM-cleaned dataset, which in my opinion would be the clearest way to demonstrate the effectiveness of the LLM-based data clean method proposed in this work.

**Reproducibility:**

3: Could reproduce the results with some difficulty. The settings of parameters are underspecified or subjectively determined; the training/evaluation data are not widely available.

**Reviewer Confidence:**

4: Quite sure. I tried to check the important points carefully. It's unlikely, though conceivable, that I missed something that should affect my ratings.

---

> ### Author Rebuttal · Authors · 2023-08-29
>
> We would like to thank you for thoroughly reading the paper and providing valuable feedback. We have carefully read your reviews and prepared our responses accordingly to address your concerns. Our responses to your questions are stated below.
>
> **Response to Reasons to Reject:**
>
> **Reason 1:** Though we already did the domain generalization evaluation on the MS-MARCO dataset to demonstrate the effectiveness of LLM-generated queries, we also put the results for the in-domain Debatepedia dataset as we found similar analysis on the in-domain dataset in the literature for datasets related issues. For instance:
>
> *Guo, Y., Clavel, C., Eddine, M. K., & Vazirgiannis, M. (2022, December). Questioning the Validity of Summarization Datasets and Improving Their Factual Consistency. In Proceedings of the 2022 Conference on Empirical Methods in Natural Language Processing (pp. 5716-5727).*
>
> *Laskar, M. T. R., Chen, C., Fu, X. Y., & Tn, S. B. (2022, December). Improving Named Entity Recognition in Telephone Conversations via Effective Active Learning with Human in the Loop. In Proceedings of the Fourth Workshop on Data Science with Human-in-the-Loop (Language Advances) (pp. 88-93).*
>
> **Reason 2:** We initially reported the results based on only the ROUGE metric as this is mostly done when the results are reported for Debatepedia [1, 2]. Moreover, since we conducted extensive experiments in different scenarios, filtered and unfiltered versions, on both Debatepedia for in-domain evaluation as well as MS-MARCO for domain generalization, conducting human evaluation on the outputs of all models will require significant time. However, based on your suggestions to conduct a qualitative evaluation, we utilize GPT-4 to conduct human-like evaluation due to its impressive performance as an evaluator across various tasks, including state-of-the-art performance on the summarization task as a human-like evaluator [3].
>
> In our human-like qualitative evaluation using GPT-4, we ask GPT-4 to score between 1 to 5 for the factual consistency and the coherence of the summaries generated by different models, as generating a factually correct summary as well as maintaining coherence during summary generation two of the important metrics that may depend on the given queries.
>
> We sample 10 summaries generated by each model on the MS-MARCO dataset and compute the average score for the filtered version and the unfiltered version (our results are stated below). We find that the factual consistency and the coherence score of the models are much better in the filtered version in comparison to the unfiltered version:
>
> CQSUMDP (filtered):
> - Coherence: 2.87
> - Factual Consistency: 3.00
>
> CQSUMDP (unfiltered):
> - Coherence: 1.60
> - Factual Consistency: 1.01
>
> PQSUMDP (filtered):
> - Coherence: 3.30
> - Factual Consistency: 2.93
>
> PQSUMDP (unfiltered):
> - Coherence: 1.27
> - Factual Consistency: 1.00
>
> Original Debatepedia (filtered):
> - Coherence: 2.87
> - Factual Consistency: 2.87
>
> Original Debatepedia (unfiltered):
> - Coherence: 1.33
> - Factual Consistency: 0.98
>
> References:
>
> [1] Preksha Nema, Mitesh M. Khapra, Anirban Laha, and Balaraman Ravindran. 2017. Diversity driven attention model for query-based abstractive summarization. In Proceedings of the 55th Annual Meeting of the Association for Computational Linguistics (Volume 1: Long Papers), pages 1063–1072, Vancouver, Canada. Association for Computational Linguistics.
>
> [2] Md Tahmid Rahman Laskar, Enamul Hoque, and Jimmy Xiangji Huang. 2022. Domain Adaptation with Pre-trained Transformers for Query-Focused Abstractive Text Summarization. Computational Linguistics, 48(2):279–320.
>
> [3] Liu, Yang, Dan Iter, Yichong Xu, Shuohang Wang, Ruochen Xu, and Chenguang Zhu. "Gpteval: Nlg evaluation using gpt-4 with better human alignment." arXiv preprint arXiv:2303.16634 (2023).
>
> **Reason 3:** We did not do that initially because such human evaluations in the original queries are already conducted in earlier research, which we cited in the paper [1] as well as specifically mentioned in lines 611-620. Based on your suggestion, we conducted human evaluation again on a randomly sampled 100 instances. But this time, we ask the annotators to select the preferred queries from Original Queries, ChatGPT-generated, and PaLM-generated. We find that the ChatGPT-generated queries were preferred 56% times, PaLM-generated ones 32% times, and the original queries 12% times.
>
> [1] Md Tahmid Rahman Laskar, Enamul Hoque, and Jimmy Xiangji Huang. 2022. Domain Adaptation with Pre-trained Transformers for Query-Focused Abstractive Text Summarization. Computational Linguistics, 48(2):279–320.
>
> **Questions for the Authors:**
>
> **What are the inter-annotator agreements and the specific setting of the human evaluation you conducted for comparing the quality of queries written by ChatGPT and PaLM?** Using Cohen’s Kappa, we obtain good agreement, 0.6 < k <= 0.8 between annotators. The annotators were required to select the preferred query based on considering the following: (i) relevance with the summary, (ii) relevance with the document, and (iii) conciseness.
>
> **Reproducibility Concern:** Regarding the reproducibility concern, we would like to inform that to ensure reproducibility, we have already included the dataset alongside the supplementary material. Moreover, we have already mentioned the computing resources we used for experiments, alongside hyperparameters and the model details, etc. in Section A.3.3 in the paper (please see lines: 701-724). Meanwhile, the dataset will be made publicly available upon acceptance of the paper.

---

### Official Review · Reviewer_8uYK · 2023-08-10

**Soundness:** 2

**Excitement:**

3: Ambivalent: It has merits (e.g., it reports state-of-the-art results, the idea is nice), but there are key weaknesses (e.g., it describes incremental work), and it can significantly benefit from another round of revision. However, I won't object to accepting it if my co-reviewers champion it.

**Paper Topic And Main Contributions:**

In this paper, the authors present a methodology for fixing queries of query-focused abstractive summarization that do not match the reference summaries of the document-query-summary samples. They exemplify this methodology with the Debatepedia dataset. ChatGPT and Palm-2 are used for regenerating the samples' queries and creating new versions of the Debatepedia dataset. Later, (BART, PEGASUS, and T5)-based query-focused abstractive summarization models are used to measure the impact on the performance after rewriting the queries with LLMs. The results show an improvement in the performance of some summarization models.

**Questions For The Authors:**

A) What are the statistics of the datasets after regenerating the queries?

B) How much do the generated queries overlap with the reference summaries? It would be interesting to measure the similarity between them.

C) Why, on the second method, did you regenerate the query for all the filtered samples instead of applying the relevance classifier and re-generate only those with a query-summary mismatch?

D) Table 1. Do CQSumDP, PQSumDP, and Original Debatepedia have the same size?

E) Tables 1 and 2 show marginal improvements over the original dataset that could not be statistically significant. Have you measured the confidence intervals of the results to validate the significance? They should be added to the tables to refute the improvements.

F) In Section 3. What kind of queries do we find in Debatepedia dataset?

G) Lines 373-379. How many queries did you check in total? Why is it not remarked in the main article content?

H) Lines 310-311 and 387-391. Why did you write "we ask three human evaluators" when you evaluated those samples? It could derivate into misleading information. Also, it could carry a certain bias.

I) Lines 309-318. What was your inter-agreement? It will be interesting to have that information.

J) Table 3, Example 1. Why do you state that the query and the summary are not relevant? The only mismatch between them seems to be MBA and PhD; the rest seems to be completely relevant.

**Reasons To Accept:**

Exploring the usage of Large Language Models for improving/fixing data in existing datasets or creating new datasets is a clear research path. This work defines a good case of study around the Debatepedia dataset, where some samples suffer from a specific problem, such as query-summary mismatching.

**Reasons To Reject:**

This work presents some weaknesses regarding the information provided:

- There is no study around LLM-based data sampling, where Chat-GPT or PaLM-2 detects which queries should be rewritten. There is no measurement of the performance of those models for that classification task.

- There is no data on the nature and quality of the generated queries.

- It is unclear whether the methodology posed is fixing (maintaining all samples and fixing some of them) or filtering and re-generating a subset of the original dataset.

- The size of the generated datasets (CQSumDP, PQSumDP) is unclear. Thus, the results presented could also be unclear.

- Superficial data results analysis.

Also, a big concern about the methodology: re-generating the queries with an LLM that simultaneously sees the document and the summary could generate queries excessively biased towards the reference summaries that could help the summarization models.

**Reproducibility:**

5: Could easily reproduce the results.

**Reviewer Confidence:**

4: Quite sure. I tried to check the important points carefully. It's unlikely, though conceivable, that I missed something that should affect my ratings.

**Typos Grammar Style And Presentation Improvements:**

Typos/Style:

- 66-67: whether SOTA LLMs can be utilized

- 68: Our experiments

- 145: we

- 147: and regenerated these queries

Suggestions:

- 225-227: You can not ask to generate a query and summary giving a document since this would be a completely (and synthetic) dataset. It is not because of the length of the summaries.

- 326-330: Having some reference that refutes this affirmation would be nice.

---

> ### Author Rebuttal · Authors · 2023-08-29
>
> We would like to thank you for kindly reviewing the paper and providing your feedback. We have carefully read your reviews and prepared our responses accordingly to address your concerns. Our responses to your comments are stated below.
>
> **Response to Reasons to Reject:**
>
> **Regarding no study around LLM-based data sampling and no measurement of the performance of those models for that classification task:** In our submitted manuscript, we already conducted extensive experiments with our two approaches, the (i) unfiltered version: where only queries that are detected as not relevant were regenerated by LLMs while keeping the rest as intact, and the (ii) filtered version: where we sample some instances based on some rules and only regenerated the queries for the sampled instances while excluding the rest. We found that LLM-generated queries led to performance gains for both approaches in comparison to the original queries in the Debatepedia dataset. Our results are presented in Table 1 and Table 2. While our findings demonstrate that classifying by LLMs is not that useful for data sampling in comparison to rule-based sampling, the query regeneration by LLMs is found to be useful for performance improvement in both cases. To also measure the capability of LLMs in classifying whether the query is relevant to the document/summary, we sample 100 instances and evaluate using humans to know if they also agree with the classification done by LLMs. We find that the precision for the classification task for PaLM-2 is 71% while for ChatGPT is 66%. For the human evaluation, usually, at least 2 annotators did the evaluation, and if there was a disagreement, another annotator chimed and we considered the majority votes.
>
> **Regarding the nature and quality of the generated queries:** We have already shown some queries generated by different LLMs in Section A.3.6 in Table 7. Note that since we submitted to the short paper track, we had to demonstrate this in the appendix. Here, we also recap our analysis of the paper.
> For our analysis, we collect a set of 3 samples from this dataset and present them in Table 7. While comparing between the queries in the first example in the table, we find that the original query is just one-word length and very ambiguous, while the ChatGPT-generated query is more descriptive and more relevant to both the document and the summary. For the second example, we find that even though the original query is descriptive, it does not have any relevance to the generated summary. Whereas both the ChatGPT and PaLM generated queries are very relevant to both the document and the summary (in this example, the PaLM generated query is more descriptive). For the third example, we find that the original query is related to “entrepreneurs”. However, the document is about “product managers”, not “entrepreneurs”. Meanwhile, the ChatGPT and PaLM generated queries are also very relevant to the document and both of these LLM-generated queries are the same. This analysis further demonstrates the relevance of our LLM-generated query in comparison to the original query in Debatepedia. For convenience, we also submitted our constructed dataset containing the regenerated queries during the paper submission which can be found in the supplementary material.
>
> **Regarding whether the methodology posed is fixing (maintaining all samples and fixing some of them) or filtering and re-generating a subset of the original dataset:** As mentioned in Section 3 of our paper, we proposed two approaches: (i) unfiltered version: maintaining all samples but only regenerate the queries by LLMs that are considered as problematic by LLMs, (ii) filtered version: exclude samples that seems irrelevant (e.g., very short documents which are even shorter than gold summaries, summaries that are more of close-ended QA: yes/no type, etc.) and only regenerate the queries by LLMs for the ones that are not excluded.
> In our submitted manuscript, we already conducted extensive experiments with these two approaches and found that LLM-generated queries led to performance gain for both approaches in comparison to the original queries in the Debatepedia dataset. While we cannot directly compare these approaches based on their performance in the Debatepedia test set since the size of the dataset for the filtered version and unfiltered version is different, we find that in the MS-Marco dataset, the performance of models trained on the filtered version is superior than the unfiltered version (please see Table 2), establishing the effectiveness of our approach that utilizes rule-based technique for data sampling followed by regenerating the queries for the sampled instances using LLMs.
>
> **Regarding the size of the generated datasets (CQSumDP, PQSumDP):** We have already mentioned this in line 214-216 of our submitted manuscript: the filtered versions size: 5212/301/401 instances in training, validation, and test sets while for the unfiltered datasets: 12000/719/1000 in training, validation, and test sets, respectively.
>
> **Regarding superficial data results analysis:** While we submitted a short paper, we already demonstrated our analysis on the in-domain dataset: Debatepedia and domain generalization on the MS-MARCO dataset. We also presented human evaluation of the LLM-Generated queries in the main paper. Meanwhile, due to the restriction of 4 pages in the short paper track, we presented the results based on model scaling, ablation studies, zero-shot learning, and qualitative analysis of the annotated dataset in the Appendix (Section A.3.3, A.3.4, A.3.5, A.3.6, lines 725 to 817). It should be noted that since the accepted papers will get one additional page, we can also demonstrate some of these findings presented in the appendix in the main paper.
>
> **Regarding whether re-generating the queries with an LLM that simultaneously sees the document and the summary could generate queries excessively biased towards the reference summaries:** Thanks for pointing this out. We have computed this for all types of queries: original queries, PaLM-generated queries, as well as ChatGPT-generated queries, and find that the frequency of words that appear both in queries and summaries are 17% when original queries were used, 17% when generated by PaLM-2, and 19% when generated by ChatGPT. Thus, LLM-generated queries did not have any significantly high overlaps with the words in the reference summaries. More importantly, we find that the LLM-based query regeneration approach, especially the one based on ChatGPT, is generally better for not only the Debatepedia dataset but also for the out-of-domain dataset: the MS-MARCO dataset. This further mitigates any concerns of bias towards the reference summaries.
>
> **Questions For The Authors:**
>
> **What are the statistics of the datasets after regenerating the queries?**
> We have mentioned this in lines 214-216 of our submitted manuscript. The filtered version size: 5212/301/401 instances in training, validation, and test sets, while the unfiltered version size: 12000/719/1000 in training, validation, and test sets, respectively.
>
> **How much do the generated queries overlap with the reference summaries? It would be interesting to measure the similarity between them.**
> We have computed this for both original queries, PaLM-generated queries, as well as ChatGPT-generated queries and find that the frequency of words that appear both in queries and summaries are 17% when original queries were used, 17% when generated by PaLM-2, and 19% when generated by ChatGPT.
>
> **Why, on the second method, did you regenerate the query for all the filtered samples instead of applying the relevance classifier and re-generate only those with a query-summary mismatch?** This is done to compare our two proposed approaches: (i) whether we can solely rely on LLMs for both relevancy classification and query regeneration, or (ii) can we sample instances via rule-based techniques designed by humans to address existing issues in Debatepedia and only re-generate the queries using LLMs for the sampled instances. So in our first approach, we have already done this: “applying the relevance classifier and re-generate only those with a query-summary mismatch?”. On the second approach, we address existing issues in Debatepedia (e.g., short documents that are even shorter than gold reference summaries, summary length of only 1-2 words that are more of close-ended question-answering, etc.) by removing instances that are less relevant for query-focused summarization and only re-generate the queries for the filtered instances.
>
> **Table 1. Do CQSumDP, PQSumDP, and Original Debatepedia have the same size?** We have mentioned this in lines 214-216 of our submitted manuscript. The filtered version size: 5212/301/401 instances in training, validation, and test sets, while the unfiltered version size: 12000/719/1000 in training, validation, and test sets, respectively. So the respective versions (filtered and unfiltered) will have the same size for different annotations (LLM and Original): CQSumDP, PQSumDP, Original Debatepedia. Thus, the filtered versions size for CQSumDP, PQSumDP, and Original Debatepedia are 5212/301/401 instances in training, validation, and test sets; while for the unfiltered datasets, the sizes for CQSumDP, PQSumDP, Original Debatepedia are: 12000/719/1000 in training, validation, and test sets, respectively.
>
> **Tables 1 and 2 show marginal improvements over the original dataset that could not be statistically significant. Have you measured the confidence intervals of the results to validate the significance? They should be added to the tables to refute the improvements.** As we submitted to the short paper track based on our early findings, unfortunately, we did not conduct any significance tests in our initial manuscript. Meanwhile, the test set of filtered and unfiltered versions for Debatepedia are different, and so we can not directly compare these two versions on Debatepedia to measure whether the performance gains are significant or not. However, since we utilize MS-MARCO for domain generalization performance evaluation, the evaluation datasets are the same in this case. Thus, we conduct significance tests for the filtered and unfiltered versions of CQSumDP, PQSumDP, and Original Debatepedia when these datasets are used for model training followed by evaluation on MS-MARCO.
> Based on paired t-test, we find that the filtered versions for each model significantly (**p ≤ .05**) outperform the unfiltered versions in terms of all ROUGE metrics. However, the difference is not statistically significant in terms of BERTScore. Note that for MS-MARCO, the ROUGE metric, especially ROUGE-L is the more standard metric, as used in the following papers:
>
> *Laskar, M. T. R., Hoque, E., & Huang, J. X. (2022). Domain adaptation with pre-trained transformers for query-focused abstractive text summarization. Computational Linguistics, 48(2), 279-320.*
>
> *Nishida, K., Saito, I., Nishida, K., Shinoda, K., Otsuka, A., Asano, H., & Tomita, J. (2019, July). Multi-style Generative Reading Comprehension. In Proceedings of the 57th Annual Meeting of the Association for Computational Linguistics (pp. 2273-2284).*
>
> **In Section 3. What kind of queries do we find in the Debatepedia dataset?** As mentioned in lines 91-95, the Debatepedia dataset requires generating summaries from a document containing argumentative text (i.e., arguments and counter-arguments) for the given query. We have also presented the limitation of existing queries in Debatepedia in Section A.1 (lines 611 to 638):
> - 52% of the queries in this dataset have no relevance to the documents or the summaries, as demonstrated in Table 3.
> - 70% of the queries are close-ended (i.e., Yes/No type) questions (see Example 4 in Table 3).
>
> **Lines 373-379. How many queries did you check in total? Why is it not remarked in the main article content?** Thanks for pointing this out. Note that we have checked all the queries in the filtered versions, consisting of 5212/301/401 queries in the training/validation/test sets, respectively. We will mention this in the main article in the next iteration of the paper.
>
> **Lines 310-311 and 387-391. Why did you write "we ask three human evaluators" when you evaluated those samples? It could derivate into misleading information. Also, it could carry a certain bias.** We will fix this accordingly. Note that the evaluation of these LLMs were done by the authors since we were only evaluating the queries generated by two of the existing LLMs that we study in this paper. We did not propose any new LLMs in this paper and so there are no risks of any potential biases while doing this evaluation. To further ensure high quality evaluation and mitigate any risks of potential biases, the human evaluators did not have any information on which queries were generated by which LLMs. Meanwhile, evaluations of LLMs via paper authors have also been done in recent months. For instance, human evaluation of LLMs by authors have been done in the following paper in ACL 2023:
>
> *Md Tahmid Rahman Laskar, M Saiful Bari, Mizanur Rahman, Md Amran Hossen Bhuiyan, Shafiq Joty, and Jimmy Huang. 2023. A Systematic Study and Comprehensive Evaluation of ChatGPT on Benchmark Datasets. In Findings of the Association for Computational Linguistics: ACL 2023, pages 431–469, Toronto, Canada. Association for Computational Linguistics.*
>
> **Lines 309-318. What was your inter-agreement? It will be interesting to have that information.** Using Cohen’s Kappa, we obtain good agreement, 0.6 < k <= 0.8 between annotators.
>
> **Table 3, Example 1. Why do you state that the query and the summary are not relevant? The only mismatch between them seems to be MBA and PhD; the rest seems to be completely relevant.** We state this because the query wanted to know whether the MBA degree can enhance specific skills but the summary rather generated something related to another degree. Thus, we consider this query-summary pair as irrelevant. While we agree that the rest seems relevant, still there are discrepancies between what was required in the query and what was in the summary.
>
> **Typos Grammar Style And Presentation Improvements:** We will fix all the typos that are stated.
>
> **Reproducibility Concern:** Regarding the reproducibility concern, we would like to inform that to ensure reproducibility, we have already included the dataset in the supplementary material. Moreover, we have already mentioned the computing resources we used for experiments, alongside hyperparameters and the model details, etc. in Section A.3.3 in the paper (please see line: 701-724). Meanwhile, the dataset will also be made publicly available upon acceptance of the paper, as mentioned in the submitted manuscript.

---

### Meta-Review · Area_Chair_W5j9 · 2023-09-19

**Recommendation:** 3

**Metareview:**

This paper investigates how we can use LLMs to clean noisy Debatepedia for query-focused summarization using rule-based filtering and query rewriting. Reviewers agree that the experiments have demonstrated the effectiveness of the proposed approach, and this could have potential impacts if we can use LLM to fix annotation errors for broader tasks. The author response has clarified the reason why it reports both in-domain and out-of-domain evaluation and additional GPT-4 based and human evaluation to show the improvements and query quality. The paper is narrow in scope which limits its excitement. In addition to the new results, we also strongly encourage authors to incorporate the feedback to improve the paper clarity.

---

### Decision · Program_Chairs · 2023-10-07

**Decision:**

Accept-Findings

**Comment:**

This paper investigates how we can use LLMs to clean noisy Debatepedia for query-focused summarization using rule-based filtering and query rewriting. Reviewers agree that the experiments have demonstrated the effectiveness of the proposed approach, and this could have potential impacts if we can use LLM to fix annotation errors for broader tasks. The author response has clarified the reason why it reports both in-domain and out-of-domain evaluation and additional GPT-4 based and human evaluation to show the improvements and query quality. The paper is narrow in scope which limits its excitement. In addition to the new results, we also strongly encourage authors to incorporate the feedback to improve the paper clarity.